# Corticothalamic phase synchrony and cross-frequency coupling predict human memory formation

Catherine M Sweeney-Reed[1]*, Tino Zaehle[1], Juergen Voges[1,2], Friedhelm C Schmitt[1], Lars Buentjen[1], Klaus Kopitzki[1,2], Christine Esslinger[1], Hermann Hinrichs[1,2,4], Hans-Jochen Heinze[1,2,4], Robert T Knight[3], Alan Richardson-Klavehn[1,2]*

[1]Departments of Neurology and Stereotactic Neurosurgery, Otto von Guericke University, Magdeburg, Germany; [2]Department of Behavioral Neurology, Leibniz Institute for Neurobiology, Magdeburg, Germany; [3]Helen Wills Neuroscience Institute and Department of Psychology, University of California, Berkeley, Berkeley, United States; [4]German Centre for Neurodegenerative Diseases, Magdeburg, Germany

**Abstract** The anterior thalamic nucleus (ATN) is thought to play an important role in a brain network involving the hippocampus and neocortex, which enables human memories to be formed. However, its small size and location deep within the brain have impeded direct investigation in humans with non-invasive techniques. Here we provide direct evidence for a functional role for the ATN in memory formation from rare simultaneous human intrathalamic and scalp electroencephalogram (EEG) recordings from eight volunteering patients receiving intrathalamic electrodes implanted for the treatment of epilepsy, demonstrating real-time communication between neocortex and ATN during successful memory encoding. Neocortical-ATN theta oscillatory phase synchrony of local field potentials and neocortical-theta-to-ATN-gamma cross-frequency coupling during presentation of complex photographic scenes predicted later memory for the scenes, demonstrating a key role for the ATN in human memory encoding.

**\*For correspondence:** catherine.sweeney-reed@med.ovgu.de (CMS-R); alan.richardson-klavehn@med.ovgu.de (AR-K)

**Competing interests:** The authors declare that no competing interests exist.

## Introduction

The anterior thalamic nuclei (ATN) are thought to play an important role in an extended hippocampal network central to memory formation (encoding) and novelty processing, which coordinates synaptic changes involving widespread neocortical areas, enabling life events to be recorded and later reinstated (*Knight, 1996*; *Aggleton et al., 2010*; *Nyhus and Curran, 2011*; *Ritchey et al., 2013*; *Schott et al., 2013*). While corticothalamic interactions are well-known to be crucial for adaptive behavior (*Saalmann et al., 2012*), the functional role of the ATN in humans has resisted investigation with non-invasive techniques owing both to its depth and small size. Here we had the rare opportunity to record electrophysiological activity during memory encoding directly from the ATN and dorsomedial thalamic nuclei (DMTN) of eight epileptic human volunteers with electrodes implanted for epilepsy treatment, as well as from frontal (and in two cases, parietal) scalp electrodes, reflecting neocortical activity (*Figure 1*). The DMTN are thought to be involved in executive control during memory retrieval (*Van der Werf et al., 2003*) and were hypothesized to play a lesser role than the ATN during encoding.

Indication that the ATN has a role in memory processing comes from human lesion and animal studies. Damage to the human ATN results in amnesia for new episodes (*Harding et al., 2000*; *Van der Werf et al., 2003*; *Aggleton et al., 2010*), and reciprocal frontal and parietal connections

**eLife digest** Memories, both the mundane and the significant, play an integral role in our daily lives. Scientists have long sought to establish exactly how our memories are formed; how does an experience, with its sights, sounds and feelings, become a mental representation stored within our brain?

One way to investigate this question is to look at the activity of different parts of the brain. Brain imaging techniques have helped researchers identify two key brain regions that are involved in the process of memory formation: the neocortex and the hippocampus. The neocortex forms the outer layer of the brain, and performs complex tasks such as decision-making and language comprehension. The hippocampus, which sits deeper within the brain, deals primarily with memory and navigation. Research has shown that memory formation depends on communication between the neocortex and the hippocampus. However, scientists suspected that additional structures located beneath the neocortex—among them, the anterior thalamic nuclei (ATN)—are also crucial for forming memories. This has been difficult to confirm as the small size of the ATN, and their location deep within the brain, make their activity almost impossible to monitor using standard brain imaging techniques.

One way reliable data can be recorded from the ATN is by inserting electrodes into the brain. Brain surgery of course cannot be carried out on healthy human participants, but occasionally an opportunity arises to study the brain activity of patients who have electrodes inserted for therapeutic purposes. For example, in cases where a patient's epilepsy does not respond to conventional treatments, electrodes may be implanted to electrically stimulate the ATN in an attempt to improve their symptoms.

Sweeney-Reed et al. asked eight volunteers to perform a memory task, and monitored the activity of each volunteer's ATN via electrodes that had already been implanted in their brain to treat epilepsy. Simultaneously, electrodes attached to the scalps of the volunteers recorded the activity of the neocortex. When a memory was successfully stored in the brain, the activity of the two regions became synchronized. This suggests that successful memory formation depends upon communication between the ATN and the neocortex.

While the involvement of the ATN in human memory formation has long been a topic of speculation, Sweeney-Reed et al. now provide direct biological evidence for its crucial role in the process. Consequently, future research into memory formation should focus upon the ATN in addition to the more familiar structures of the neocortex and the hippocampus.

with the ATN (*Aggleton, 2012*) have led to the hypothesis that ATN-hippocampal connections play a regulatory role in encoding (*Vertes et al., 2001*; *Aggleton et al., 2010*). Notably, 75% of ATN oscillatory power in non-human animal studies is in the theta (4–8 Hz) range, the dominant hippocampal rhythm (*Vertes et al., 2001*), which is also implicated in ATN-hippocampal communication (*Vertes et al., 2001*; *Aggleton et al., 2010*).

Neural communication and synaptic plasticity rely on long-range phase–phase and phase–amplitude synchrony of neural oscillations (*Lachaux et al., 1999*; *Lisman and Jensen, 2013*). Fronto-hippocampal theta synchrony accompanies memory formation (*Benchenane et al., 2010*; *Nyhus and Curran, 2011*). Furthermore, theta synchrony is reported to bind medial temporal (MT) and parietal areas during associative encoding (*Crespo-Garcia et al., 2010*), and MT with frontal and parietal cortex during successful retrieval (*Watrous et al., 2013*). Neocortical and hippocampal gamma (>30 Hz) oscillations appear to reflect local processing related to activation and maintenance of neuronal object representations (*Jensen et al., 2007*), binding diverse perceptual and contextual information (*Nyhus and Curran, 2011*). Spatially separate gamma oscillations have been found to be locked to the phase of the theta oscillation, supporting binding of the coherent ensemble underlying a given memory trace (*Mizuhara and Yamaguchi, 2011*). Indeed such gamma-power-to-theta-phase cross-frequency coupling (CFC) has been identified within human neocortex during word recognition (*Canolty et al., 2006*), working memory maintenance (*Axmacher et al., 2010*), and in successful long-term memory encoding (*Friese et al., 2012*), as well as within the rat hippocampus (*Colgin et al., 2009*), and has been proposed as a mechanism for transiently coupling distributed cortical activity (*Canolty et al., 2006*; *Friese et al., 2012*; *Lisman and Jensen, 2013*).

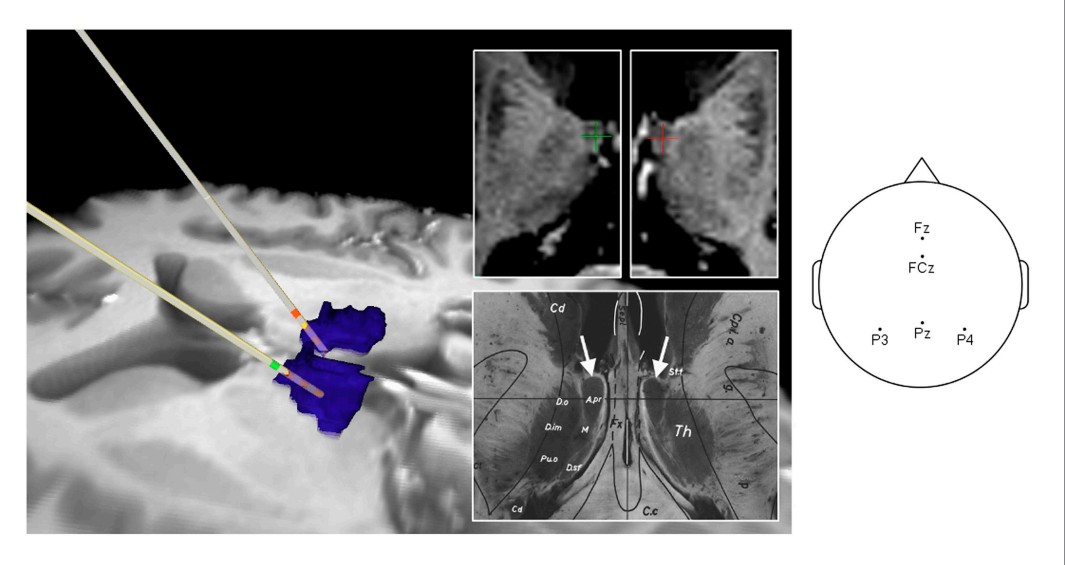

**Figure 1**. Intracranial electrode location in Participant 1. Left: Reconstruction of intrathalamic contact location using intraoperative X-ray image coordinates, superimposed on preoperative MRI scan. Dorsomedial thalamic nucleus (DMTN): blue (localized using masks from Wake Forest University Pick Atlas [http://fmri.wfubmc.edu/software/PickAtlas (*Maldjian et al., 2003*)] warped into participant's brain space). Anterior thalamic nucleus (ATN) contacts: green (left), red (right). Middle: Most superficial contacts (upper panel) clearly lie in the ATN, by reference to Schaltenbrand atlas (*Schaltenbrand and Wahren, 1977*) (lower panel: A. pr. = nucleus anterior principalis). Right: Scalp electrode locations for this participant. Lower panel reproduced from G Schaltenbrand and W Wahren, *Atlas for Stereotaxy of the Human Brain* (1977), published by Thieme Medical. The image is used here with permission from the copyright holders, who retain the copyright. All rights reserved. Please refer to the original.

Together, these findings suggest that neocortical-ATN communication might be related to local ATN processing during encoding. We hypothesized that neocortical-ATN theta phase synchrony, and the relationship between theta phase and local ATN gamma amplitudes, would be critical for memory encoding.

## Results

We assessed the role of the ATN in memory encoding by contrasting electrophysiological activity during successful compared with unsuccessful encoding of serially presented photographs of 200 complex indoor and outdoor scenes. Participants judged whether each photograph depicted an indoor or an outdoor scene. Successful encoding was defined as correct recognition of a scene as old on a subsequent recognition memory test combining photographs of old and similar new scenes. All eight participants were able to discriminate old from new scenes (*Table 1*). Mean successful encoding across participants was 55%, and mean unsuccessful encoding was 45%, resulting in means of 101 and 87 observations per category, respectively, for EEG analysis.

All significance tests reported here were two-tailed. Response times for the indoor/outdoor judgment at encoding were compared between successfully (group mean = 1.04 s) and unsuccessfully (group mean = 1.14 s) encoded scenes for the seven participants for whom the paradigm was identical (see 'Materials and methods'), and no difference was detected (paired T-test: T = 1.81, p = 0.12, with 6° of freedom, DF). For each participant, the difference between their expected probability of sequential successful encoding, calculated according to their overall rate of successful encoding, and their observed probability of sequential successful encoding, was calculated. The mean difference between these probabilities across the group did not significantly differ from zero (one-sample T-test: T = 1.35, p = 0.22, with 7 DF). Both these behavioral findings suggest that the neural findings to be reported did not reflect simple global attentional fluctuations (see 'Discussion').

We contrasted key oscillatory features of the EEG (512 Hz sampling frequency) during successful compared with unsuccessful encoding. To assess long-range communication, we calculated an

**Table 1.** Behavioral results

| Pt | SE % (NE) | SE-C % | UE % (NE) | FA % (NE) | FA-C% | CR % (NE) | SE-FA % | SE-C–FA-C % |
|------|-----------|--------|-----------|-----------|-------|-----------|---------|-------------|
| 1 | 74 (74) | 74 | 26 (26) | 29 (29) | 25 | 71 (71) | 45 | 49 |
| 2 | 66 (132) | 39.5 | 34 (68) | 36 (72) | 6.5 | 64 (128) | 30 | 33 |
| 3 | 42.5 (85) | 35.5 | 57.5 (115) | 5 (10) | 1.5 | 95 (190) | 37.5 | 34 |
| 4 | 36.5 (73) | 27.5 | 63.5 (127) | 21 (42) | 6.5 | 79 (158) | 15.5 | 21 |
| 5 | 57.5 (115) | 38 | 42.5 (85) | 47 (94) | 13.5 | 53 (106) | 10.5 | 24.5 |
| 6 | 38.5 (77) | 36.5 | 61.5 (123) | 10 (20) | 4 | 90 (180) | 28.5 | 32.5 |
| 7 | 81.5 (163) | 75.5 | 18.5 (37) | 63 (126) | 28.5 | 37 (74) | 18.5 | 47 |
| 8 | 43.5 (87) | 41.5 | 56.5 (113) | 25 (50) | 10 | 75 (150) | 18.5 | 31.5 |
| Mean | 55 (101) | 46 | 45 (87) | 29.5 | 11.9 | 70.5 | 25.5 | 34.1 |
| SD | 17.3 | 18.2 | 17.3 | 19.1 | 9.9 | 19.1 | 11.8 | 9.3 |

Pt = Participant. SE = successful encoding (hits). NE = number of epochs. SE-C = correctly judged 'old' with high confidence. UE = unsuccessful encoding (misses). FA = false alarms. FA-C = incorrectly judged 'old' with high confidence. CR = correct rejections. SD = standard deviation. SE-FA = index of ability to discriminate between old and new test items (i.e., hits minus false alarms). SE-C–FA-C = discrimination index for items confidently judged old.

amplitude-independent measure of phase synchrony (*Lachaux et al., 1999*) of oscillations in local field potentials recorded from thalamus and neocortex. We then employed CFC to assess the relationship between long-range communication and local neural processing, and Granger causality (GC) to assess likely direction of influence.

Mean fronto-thalamic phase-locking values (PLVs) across all eight participants during encoding are plotted against time and frequency in *Figure 2*. Corticothalamic theta synchrony differences in a late (0.5–1.5 s) time window were predicted, given the theoretical importance of theta oscillations in memory, and the typical late timing of encoding-related differences in frontal and parietal event-related potentials (ERPs) (*Schott et al., 2002*) and in post-stimulus MT theta oscillatory power (*Hanslmayr and Staudigl, 2014*). Frontal-right-ATN (RATN) theta synchrony was indeed greater during successful than unsuccessful encoding at 5–6 Hz between 0.5 and 1.5 s after picture presentation (permutation tests, PT: p = 0.001; paired T-tests, TT: T = 9.9, p = 0.000022; *Figure 2*, *Figure 2—figure supplement 1*). Conservative false discovery rate correction (PT; 57 frequencies, 0–100 Hz; 1024 data-points, 0–2 s) yielded a threshold of p = 0.0019 (overall criterion p = 0.05). Cluster-size PT (CSPT) on binary TT outcomes (significant/nonsignificant at p = 0.05) showed the second cluster of this late theta synchrony to be significant (~1.0–1.5 s; p = 0.016; observed contiguous cluster 437 pixels; criterial cluster 285 pixels for overall p = 0.05). Synchrony at 5.2 Hz, averaged from 0.5 to 1.5 s for each participant, was greater during successful than unsuccessful encoding in seven of eight participants (*Figure 2E*), yielding a group difference in median synchrony (Wilcoxon test: p = 0.038). Participant 4 (*Figure 2E*) also showed the difference from 1 to 1.5 s (*Figure 2F*, *Figure 2—figure supplement 2*), so that all eight participants showed a difference in this time-window. The time course of theta synchrony is shown in *Figure 2—figure supplement 3*. Compared with surrogate data (*Theiler et al., 1992*), the PLV was significant (criterion p = 0.05) for 230 ms, which is more than a complete theta cycle at 5 Hz. The timescale is of the order of that over which synchrony is commonly detected (*Varela et al., 2001*). Note from *Figure 2F* and *Figure 2—supplement 2* that theta synchrony was greater during successful than unsuccessful encoding in all eight participants from 1.25–1.5 s post-stimulus, and the appearance of two separate episodes at the group level is likely to have arisen due to inter-participant differences in the timing of the synchrony episode over the 1 s time window from 0.5–1.5 s post-stimulus.

To further test frequency-specificity, a two-way repeated measures analysis of variance was applied to frontal-RATN PLVs obtained using theta (5.2 Hz) and beta (17.5 Hz) wavelets averaged from 0.5 to 1.5 s for each participant during successful and unsuccessful encoding. The mean PLVs showed a significant interaction between frequency and encoding success (p < 0.001). This interaction remained significant when taking adjacent theta (4.9 Hz and 5.5 Hz) and beta (16.5 Hz and 18.6 Hz) wavelets, and taking theta (5.2 Hz) and alpha (11.7 Hz) wavelets (all ps < 0.002). In all cases the advantage for successful over unsuccessful encoding for theta was greater than for alpha or beta, which showed negligible differences. The interaction was not significant when taking two theta wavelets (4.9 Hz and

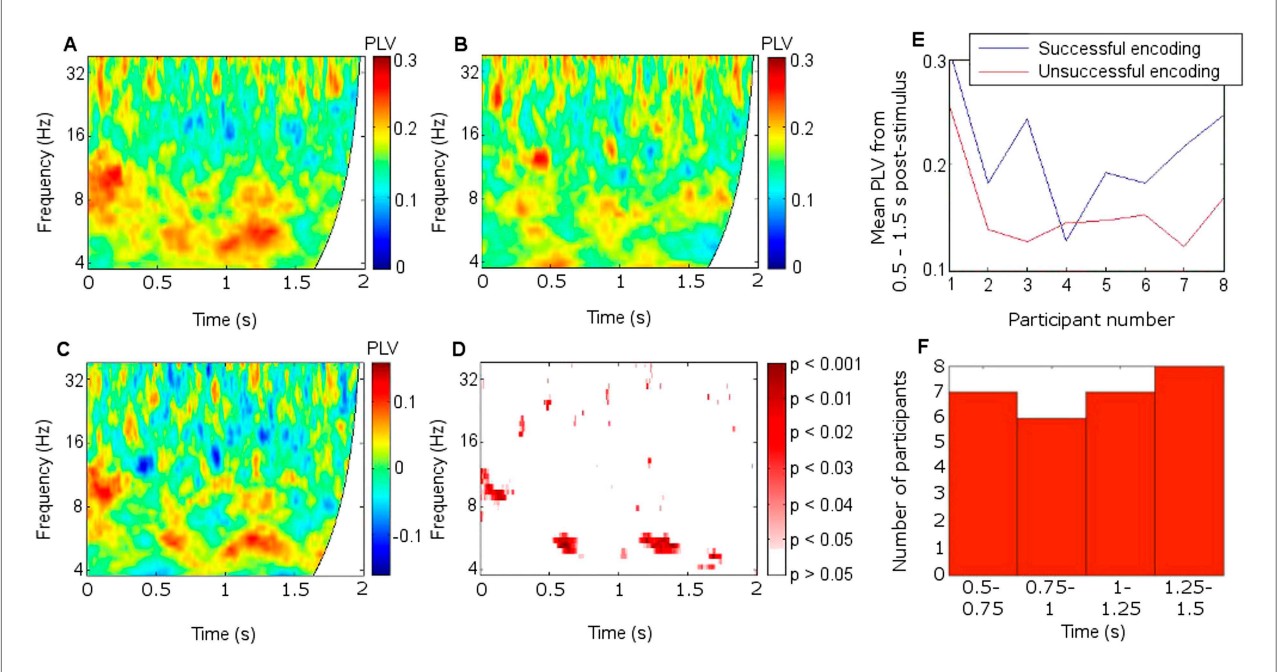

**Figure 2**. Frontal-right anterior thalamic nucleus (RATN) phase synchrony. PLV = phase-locking value. (**A**) Successful encoding. (**B**) Unsuccessful encoding. (**C**) Successful minus unsuccessful encoding. (**D**) Permutation tests: Successful minus unsuccessful encoding. (**E**) Mean theta (5.2 Hz) PLVs for successful encoding and unsuccessful encoding averaged from 0.5 to 1.5 s for the eight individual participants. (**F**) Number of participants showing greater theta synchrony during successful compared with unsuccessful encoding in four sub-time-windows from 0.5 to 1.5 s.

The following figure supplements are available for figure 2:

**Figure supplement 1**. Significance of phase synchrony between frontal neocortex and right anterior thalamic nucleus (RATN).

**Figure supplement 2**. Mean theta (5.2 Hz) phase-locking value (PLV) between frontal neocortex and right anterior thalamic nucleus (RATN) over four consecutive time windows from 0.5 to 1.5 s.

**Figure supplement 3**. Time course of theta phase synchrony.

**Figure supplement 4**. Frontothalamic synchrony involving other thalamic nuclei.

**Figure supplement 5**. Synchrony using 1-cycle wavelets to enhance time resolution.

**Figure supplement 6**. Corticothalamic phase synchrony in Participant 1.

**Figure supplement 7**. Corticothalamic phase synchrony in Participant 2.

**Figure supplement 8**. Power difference between successful and unsuccessful encoding.

**Figure supplement 9**. Synchrony with differing cortical sites.

5.2 Hz: $p > 0.30$; 5.2 Hz and 5.5 Hz: $p > 0.60$). No significant difference was observed between delta (2–4 Hz) phase synchrony during successful compared with unsuccessful encoding.

We did not observe corresponding significant synchrony differences for DMTN or left ATN (*Figure 2—figure supplement 4*). Indeed, on direct comparison, the difference in theta (5.2 Hz) PLVs between successful and unsuccessful encoding in the RATN 0.5–1.5 s post-stimulus was greater than the difference in the right DMTN (Wilcoxon test: $p = 0.018$), supporting nucleus specificity. A direct comparison of the relevant difference in theta PLVs between RATN and LATN only approached significance (Wilcoxon test: $p = 0.130$). While we do not wish to make strong claims about laterality,

the finding that the synchrony difference in RATN was significant, whereas that in LATN was not (*Figure 2—figure supplement 4*), accords well with the non-verbal scene stimuli used (*Maillard et al., 2011*) (see also 'Discussion').

There was additional early upper theta and alpha synchrony during successful compared with unsuccessful encoding (~8–12 Hz; 0–0.2 s; PT: p = 0.002; TT: T = 5.6, p = 0.00085; CSPT: p = 0.029; observed contiguous cluster 342 pixels; *Figure 2*, *Figure 2—figure supplement 1*), which might reflect enhanced item-specific attention and perception during successful encoding (*Düzel et al., 2005*). All synchrony patterns were post-stimulus (*Figure 2—figure supplement 5*). There was a significantly greater synchrony difference between successful and unsuccessful encoding in upper theta/alpha post-stimulus (0–0.2 s) than in the same frequency range during the 1 s pre-stimulus period (Wilcoxon test: p = 0.039). Furthermore, the difference between pre-stimulus theta synchrony preceding successful compared with unsuccessful encoding was not significant (Wilcoxon test: p = 0.46), whereas post-stimulus, the synchrony difference between successful and unsuccessful encoding was significant (Wilcoxon test: p = 0.039).

Because parietal scalp signals could be recorded from only two participants due to post-operative dressing placement, these were also analyzed as individual cases (*Figure 2—figure supplements 6–7*). Parietal-RATN and frontal-RATN (consistent with the group data) theta synchrony 0.5–1.5 s post-stimulus were significantly enhanced individually during successful compared with unsuccessful encoding (permutation tests on individual epochs within participants: ps < 0.05).

Because the main synchrony findings occurred in theta between 0.5 and 1.5 s, further analyses focussed on this time-frequency range. CFC was greater during successful than unsuccessful encoding between frontal theta phase-troughs and RATN gamma (~40–50 Hz) amplitude-peaks (TT, T = 4.9, p = 0.0018; CSPT: p = 0.038; observed contiguous cluster 14 pixels; criterial cluster 12 pixels for overall p = 0.05; *Figure 3*). The difference was absent for thalamic high gamma (up to 256 Hz) (*Canolty et al., 2006*).

In contrast to the frontal-RATN findings, within the RATN, theta phase-peaks were coupled with gamma amplitude-peaks. Of note is that the gamma range was narrower (~40–50 Hz compared with ~40–70 Hz) during successful than during unsuccessful encoding, resulting in significantly greater within-RATN CFC involving higher gamma-frequency amplitudes during unsuccessful compared with during successful encoding (paired T-test, TT, T = 3.6, p = 0.0086; cluster-size permutation test, CSPT: p = 0.045; observed contiguous cluster 15 pixels; criterial cluster 14 pixels for overall p = 0.05; *Figure 3—figure supplement 1A–C*). The CFC patterns during successful and unsuccessful encoding were then compared directly. Within-RATN CFC during successful encoding was compared with a distribution of 1000 phase-scattered surrogates, and the theta-gamma (3.8–8.1 Hz, 30.5–64.6 Hz) cross-frequency points at which CFC during successful encoding exceeded a threshold with criterion p = 0.05 were identified. The same analysis was performed for unsuccessful encoding. The two resulting CFC patterns differed significantly (two-dimensional Kolmogorov–Smirnov test—2-D KS test: d = 0.81, p = 0.048), with a wider gamma range during unsuccessful encoding. Assuming that an assembly of synchronously firing neurons is associated with a particular memory trace, the narrower RATN gamma range coupled with theta phase during successful memory formation could be interpreted as reflecting firing of only relevant neural assemblies, thus reflecting neural specificity during encoding (*Desimone, 1996*; *Düzel et al., 2005*; *Schott et al., 2006*). We correspondingly postulate that the corticothalamic coupling may coordinate the firing of particular thalamic neural assemblies underpinning the memory to be encoded, facilitating synaptic strengthening and relevant memory formation. Indeed, ongoing CFC has been detected in the centromedian thalamic nucleus during cognitive task performance (*Fitzgerald et al., 2013*). Again, there was no encoding-related CFC difference apparent for thalamic high gamma (*Canolty et al., 2006*).

CFC findings within the frontal cortex (*Figure 3—figure supplement 1D–F*) were consistent with the literature. Theta-to-gamma CFC was peak-to-peak at higher theta frequencies, and occurred for both successful and unsuccessful encoding (phase-scattered surrogate-data tests for successful and for unsuccessful encoding: p = 0.001). There was no significant relationship to encoding success (TT, p > 0.05; CSPT: p > 0.05; *Figure 3—figure supplement 1D–F*), consistent with hippocampal CFC patterns during working memory (*Axmacher et al., 2010*). By contrast, theta-to-gamma CFC was trough-to-peak at lower theta frequencies, and was greater during successful than unsuccessful encoding (TT, T = 3.6, p = 0.0087; CSPT: p = 0.017; observed contiguous cluster 19 pixels; criterial cluster 11 pixels for overall p = 0.05), consistent with (*Canolty et al., 2006*).

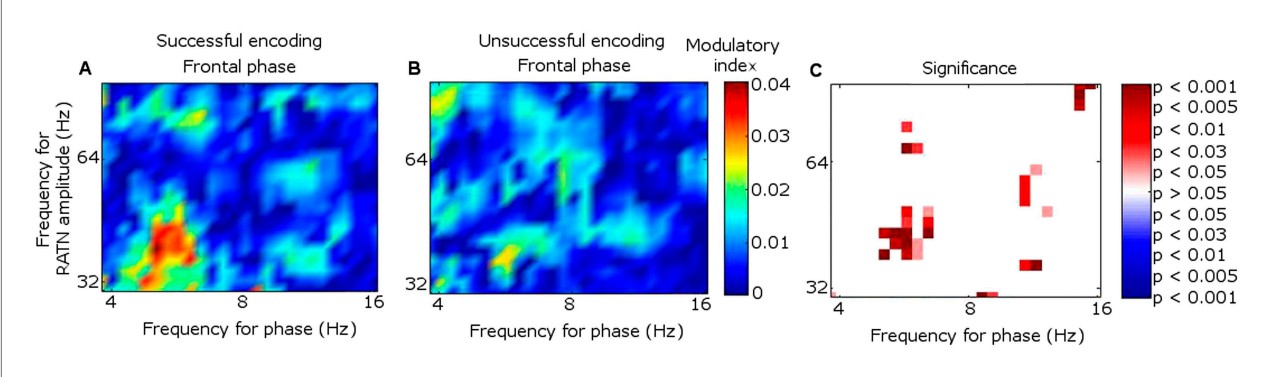

**Figure 3**. Frontal-right anterior thalamic nucleus (RATN) cross-frequency coupling (CFC). (**A**) Successful encoding. (**B**) Unsuccessful encoding. (**C**) Paired T-tests: Successful minus unsuccessful encoding.
The following figure supplements are available for figure 3:

**Figure supplement 1**. Cross-frequency coupling (CFC) within right anterior thalamic nucleus (RATN) and within frontal neocortex.

**Figure supplement 2**. Cross frequency coupling (CFC) during successful encoding in Participant 7.

**Figure supplement 3**. Cross-frequency coupling (CFC) with differing frontal cortical sites.

Theta-gamma CFC patterns within RATN and within frontal neocortex differed from the frontal-RATN pattern (*Figure 3—figure supplements 1–2*). Frontal-RATN theta-gamma (3.8–8.1 Hz, 30.5–64.6 Hz) coupling during successful vs unsuccessful encoding was compared using the above paired T-tests, to provide the cross-frequency pattern, and the same was performed for within-RATN CFC. These CFC patterns differed significantly (2-D KS test: d = 0.83, p = 0.024). Frontal-RATN and within-frontal CFC patterns also differed significantly (2-D KS test: d = 0.98, p = 0.011).

GC revealed that frontal theta better predicted RATN theta than vice-versa (*Figure 4*). In GC analyses, the peak model order provides an indication of how far into the past one signal provides information about another, and may thus be interpreted as indicating the approximate delay in transfer of information (*Staudigl et al., 2012*). We found that frontal theta prediction of RATN theta peaked at a model-order of 32, which corresponds to 63 ms, or one third of a theta cycle phase-lag between frontal and RATN theta. Such a delay is broadly consistent with frontal-RATN theta-to-gamma CFC being trough-to-peak (*Figure 3*), and intra-RATN theta-to-gamma CFC being peak-to-peak (*Figure 3—figure supplement 1A–C*), illustrated in *Figure 3—figure supplement 2*. Together, the CFC and GC findings suggest that frontal theta modulates RATN gamma via frontal-ATN theta synchrony during successful encoding.

## Discussion

We demonstrate increased corticothalamic synchrony during successful memory encoding, recording directly from the two most memory-relevant thalamic nuclei (ATN and DMTN). Our amplitude-independent phase synchrony measurements (*Lachaux et al., 1999*) show that timing of post-stimulus ATN theta activity alone (see also *Figure 2—figure supplement 8*), and its relation to local ATN processing as indexed by gamma amplitudes, is critical in successful memory encoding, providing the first electrophysiological evidence concerning the role of ATN in human memory formation. The absence of neocortical-DMTN synchrony differences, together with their recent detection during verbal memory retrieval (*Staudigl et al., 2012*), fits with evidence suggesting ATN specialization for encoding (*Harding et al., 2000*; *Van der Werf et al., 2003*), and DMTN for retrieval (*Harding et al., 2000*; *Van der Werf et al., 2003*; *Aggleton et al., 2010*; *Aggleton, 2012*). RATN involvement is moreover consistent with the non-verbal scene stimuli employed (*Maillard et al., 2011*), and with evidence that left thalamus lesions produce more severe memory deficits for verbal than non-verbal material (*Squire et al., 1989*). The late theta synchrony is in accord with the timing of differences in ERPs (*Schott et al., 2002*) and post-stimulus MT theta power (*Hanslmayr and Staudigl, 2014*) during successful compared

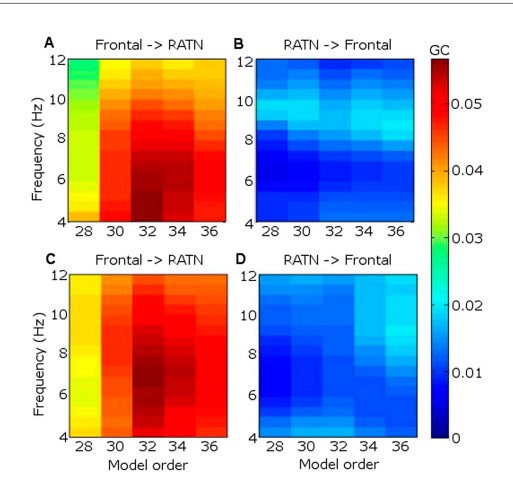

**Figure 4**. Granger causality (GC) in the theta frequency range. (**A**) During successful encoding, frontal theta activity predicted right anterior thalamic nucleus (RATN) activity, peaking at model order 32, corresponding with a 63 ms phase lag (i.e., one third of a theta cycle). (**B**) During successful encoding, RATN activity was significantly (paired T-tests: T = 3.2, p = 0.014) less predictive of frontal activity. (**C**) During unsuccessful encoding, frontal theta activity predicted RATN activity. (**D**) During unsuccessful encoding, RATN was significantly (paired T-tests: T = 3.0, p = 0.018) less predictive of frontal activity.

with unsuccessful encoding of episodic memories. While oscillations in the delta frequency range have recently been found to show a memory-related difference in the hippocampus (*Watrous et al., 2011*; *Lega et al., 2014*), we did not detect a significant difference between corticothalamic delta phase synchrony during successful compared with unsuccessful encoding.

While we provide here novel evidence that the coordination of theta and gamma oscillations involving the ATN plays a critical role in trial-by-trial memory ability, it should be noted that memory encoding is recognized to include not only content processing and information storage, but also attention (reviewed by *Kim, 2011*). Item-specific attention indeed has a well-recognized effect on whether an item is successfully encoded (*Raz and Buhle, 2006*; *Muzzio et al., 2009*; *Kim, 2011*; *Burke et al., 2014*). Attention is not only necessary for optimal memory encoding, but multiple brain structures involved in memory formation may be subject to attentional modulation, including the hippocampus itself (*Muzzio et al., 2009*). There are several indicators, however, that our findings reflect memory encoding processes beyond simply global arousal or attention fluctuations.

Firstly, the timing of the key memory differences here at around 1 s post-stimulus is strongly suggestive of a memory-related difference. Neural encoding differences related to later memory have been identified using ERPs and cortical theta/gamma oscillations, generally peaking at around 1 s after stimulus presentation, continuing up to 1.5 s post-stimulus (*Paller et al., 1987*; *Schott et al., 2002*; *Sederberg et al., 2003*; *Osipova et al., 2006*; *Lega et al., 2012*; *Hanslmayr and Staudigl, 2014*; *Long et al., 2014*). Secondly, whereas oscillatory dynamics between the pulvinar thalamic nucleus and the parietal cortex support visual attention in macaques, consistent with attention deficits following focal pulvinar lesions (*Saalmann et al., 2012*), our main findings pertain to the ATN, for which lesion and animal studies suggest a role in memory. Moreover, based on current knowledge about the ATN and the DMTN (*Van der Werf et al., 2003*) one would rather expect attentional differences to be reflected in DMTN activity, which we did not find. Most critically, if global attentional fluctuations were responsible for our theta synchrony findings, one would expect longer response times during the encoding phase for later forgotten compared with later remembered scenes, due to lack of attention to the encoding task. However, response times did not differ. Finally, global attentional fluctuations over time would also imply a dependency between the probability of successful encoding on successive trials during the study phase of the experiment, and we found no dependency.

We note, however, that variability in factors such as attention and the emotional valence of stimuli are fundamental to the study design in subsequent memory paradigms, affecting the probability of successfully encoding each item and thus enabling comparison of successful with unsuccessful encoding. Indeed, the early upper theta and alpha synchrony immediately following stimulus presentation during successful memory encoding is likely to reflect enhanced item-specific attention and perception (*Düzel et al., 2005*), with the later theta synchrony timing fitting well with previous findings relating to memory encoding (*Schott et al., 2002*; *Hanslmayr and Staudigl, 2014*).

The ATN is the target of stimulation in the treatment of focal epilepsy on the basis that seizure activity starting focally is propagated through this site to widespread cortical areas (*Lega et al., 2010*). Regional cortical specificity in frontal-RATN synchrony would support our argument that the differences that we have identified are memory-specific. While the phase synchrony pattern we report is

discernible in all three electrode placements (frontopolar, other frontal, and parietal), it is indeed not identical across regions (*Figure 2—figure supplement 9*). We also show the difference between fronto-RATN theta-gamma CFC involving the different frontal electrode placements (*Figure 3—figure supplement 3*). Again, the patterns are similar but not identical. The limited available scalp electrode coverage in our participants, however, precludes drawing strong conclusions in this regard. We note also, though, that both frontal and parietal cortices are well-recognized as being involved in memory processing (*Weiss and Rappelsberger, 2000*; *Otten et al., 2002*; *Sauseng et al., 2005*; *Uncapher and Wagner, 2009*; *Friese et al., 2012*; *Sweeney-Reed et al., 2012*).

The encoding and retrieval tasks were chosen for their simplicity, because recording was only possible at the bedside in the few days following intracranial surgery, and because we did not wish to lose data owing to failure of these rare participants to succeed at the tasks. Under these circumstances, we were able to achieve large and comparable behavioral trial numbers for successful and unsuccessful encoding, which could then be submitted to oscillatory analysis. Pilot testing revealed that one participant (not included in the current cohort) could not adequately perform an objective source-memory task under these post-operative conditions, and we thus additionally judged it unlikely that the participants could properly implement the more complex 'remember/know/guess' instructions (*Gardiner and Richardson-Klavehn, 2000*) necessary to obtain reports of subjective recollection and familiarity during recognition. Moreover, given the nature of the data ultimately obtained from our eight participants, we elected to collapse the data across the levels of response certainty to maximize trial numbers in each category for electrophysiological synchrony analysis, in a pragmatic trade-off between the power of the electrophysiological analyses and behavioral/psychologigal resolution.

Thus, a limitation of the study is that we cannot conclusively link our results to recollection, and therefore episodic memory formation, separately from familiarity. Our findings concerning ATN electrophysiological involvement in human memory formation are, however, consistent with the theoretical proposal that the ATN is a part of an extended hippocampal system supporting episodic recollection (*Aggleton, 2012*), whether during encoding or retrieval, and with the competing theoretical proposal that the ATN is specialized for encoding rather than retrieval, whether the information involved is episodic (recollection) or semantic (*Van der Werf et al., 2003*). The latter proposal is consistent with human lesion data, showing that sufferers from Korsakoff's syndrome with lesions to the ATN fail to acquire new semantic as well as new episodic information (for example, *Harding et al., 2000*) and is thus particularly consistent with our new data. We note, furthermore, that whether recollection and familiarity are separate processes (*Eichenbaum et al., 2008*), or reflect a single process, with familiarity and recollection reflecting different degrees of memory strength (*Wixted and Squire, 2008*), is an area of continuing debate (see also *Gardiner and Richardson-Klavehn, 2000* and *Yonelinas, 2002*) for relevant information regarding humans. Despite this lack of behavioral/psychological resolution, our data nevertheless provide novel evidence concerning the real-time role of the human ATN in memory formation.

It should also be noted that memory processing may be divided into different subsystems, with different components of memory processing involving different thalamic nuclei (*Mennemeier et al., 1992*). For example, whereas human lesion and animal studies suggest a regulatory role for the ATN in memory encoding (*Harding et al., 2000*; *Vertes et al., 2001*; *Van der Werf et al., 2003*; *Aggleton et al., 2010*; *Aggleton, 2012*) and for the DMTN in retrieval (*Van der Werf et al., 2003*; *Staudigl et al., 2012*), other thalamic nuclei have also been found to be involved in different aspects of memory processing. For example, evidence supports a role for the nucleus reuniens in fear conditioning and memory generalization (*Vertes et al., 2007*; *Xu and Südhof, 2013*), as well as spatial processing (*Jankowski et al., 2014*), for the pulvinar nucleus in attention (*Saalmann et al., 2012*) and nonverbal memory processing (*Johnson and Ojemann, 2000*), and for the left ventro-lateral thalamus in verbal memory encoding (*Johnson and Ojemann, 2000*). Our focus on the ATN and DMTN is based on the rare availability of human electrophysiological data from these sites, which was determined by clinical requirements, and is consistent with extant data concerning the amnesic effects of lesions in these thalamic areas in humans (*Harding et al., 2000*; *Van der Werf et al., 2003*).

In summary, our findings shed new light on real-time interaction between the ATN and the neocortex, thus broadening understanding of the brain structures involved in memory formation from a focus on the hippocampus and neocortex to recognition of a pivotal role for the ATN. More generally, the CFC theta-gamma findings, together with the amnesic effects of lesions to the human ATN,

provide evidence that the ATN plays an active role in encoding, instead of simply relaying cortical or hippocampal signals (*Aggleton et al., 2010*).

## Materials and methods

### Participants, and intracranial and scalp recording

Intrathalamic data were recorded from 1.5 mm platinum-iridium electrodes implanted bilaterally (four contacts each) in the thalamus for stimulation therapy for multiple pharmacoresistant focal epilepsy in eight adult participants, all of whom were not suitable candidates for resective surgery. A minimum sample size was not set, because memory-dependent phase synchrony has been detected on an individual level in intracranial recordings (*Fell and Axmacher, 2011*; *Staudigl et al., 2012*) (see also *Figure 2*, *Figure 2—figure supplements 6–7*). The final number of participants was determined by the number of patients available in the days just following implantation during the approximately 2 years of the relevant therapeutic program, and also willing to participate in the study. This sample size was considerably greater than that usually available for intracranial studies in humans (*Canolty et al., 2006*; *Fitzgerald et al., 2013*; *Bonini et al., 2014*), and in non-human primates (*Saalmann et al., 2012*). The mean age of the participants was 37.5 years (range 28–52 years, standard deviation 8.2 years), and four participants were female. Stimulation via intracranial electrodes did not occur during the data recording. The measurements were approved by the Ethics Commission of the Medical Faculty of the Otto-von-Guericke University, Magdeburg (application number 0308), and all participants gave written informed consent in accordance with the Helsinki Declaration of 1975, as revised in 2000 and 2008. Consent to participate in our study, as well as for publication of results in an anonymized format, was obtained by the neurosurgeon at the same time as consent was obtained for the surgical procedure.

Contacts were located in the ATN for all eight participants and in the DMTN for seven of the participants (with Participant 5 not having DMTN contacts). Placement of the thalamic electrodes was performed stereotactically. The angle of entry through the skull and the depth of each electrode was calculated based on MRI images of each patient's brain pre-operatively. An intra-operative X-ray and postoperative CT-scans were carried out in order to confirm correct localization of each electrode, by reference to the Schaltenbrand and Pick Atlases (*Schaltenbrand and Wahren, 1977*; *Maldjian et al., 2003*). Scalp EEG data were simultaneously collected from frontal electrodes (Fz, AFz, or Fpz) in all eight participants, and from parietal electrodes only in Participant 1 (P3, Pz, P4; *Figure 1*, *Figure 2—figure supplement 6*) and Participant 2 (Pz only; *Figure 2—figure supplement 7*). Positioning of post-operative dressings meant that parietal electrodes could not be placed for the other six participants. Frontal electrodes were centered over Fz (frontal), AFz (anterior frontal), and Fpz (frontopolar), with the data from the most frontal available electrode from each participant included in the group analyses. These scalp sites reflect underlying cortical activity from midline frontal cortices (*Paller et al., 1987*). Data from the two participants with parietal electrodes were analyzed as single cases (*Figure 2—figure supplements 6–7*), as well as contributing to the group analyses for frontal electrodes.

Nose-referenced voltages were amplified with a Walter Graphtek (Lübeck, Germany) EEG amplifier and recorded with a 512 Hz sampling frequency. Offline re-referencing of thalamic voltages to the next deepest contact rendered six bipolar channels. Radiological intracranial electrode localization was performed for all participants intra- and postoperatively by the neurosurgeons and a physicist. Electrode location is provided in detail for Participant 1 for illustration purposes (*Figure 1*). The deepest two thalamic contacts were located in the DMTN, and the most superficial in the ATN. The second most superficial was at the ATN/DMTN border. The critical results from this participant (*Figure 2—figure supplement 6*) were obtained from a bipolar recording from the two most superficial right-sided thalamic contacts. We confirmed that the synchrony patterns were not observed when the second most superficial contact was referenced to either of the two deeper contacts, suggesting that the critical signals originated from the most superficial contact, which was clearly located in ATN, and not from its reference contact.

Note that no measureable temporal phase-shifts that simply reflect the physical (e.g., capacitive) properties of the brain-to-scalp-electrode interface have been detected at typical EEG frequencies (*Nunez and Srinivasan, 2006*). Any phase-lags between the scalp and intrathalamic recordings should, therefore, reflect genuine neural conduction delays between neocortex and thalamus. Such phase-lags

should not in any case affect the corticothalamic phase synchrony measurements detailed below, which were independent of phase-lag as well as amplitude, but were taken into account in the Granger causality calculations detailed below.

## Experimental paradigm

During encoding, the participants viewed a series of 200 (100 for Participant 1) photographs of unfamiliar real-world scenes on a computer screen. Each scene was shown for 2.4 s, followed by a fixation cross for 1.4 s, and the participant judged whether the depicted scene was indoors or outdoors (average response time around 1 s after scene presentation). The responses were made using left and right index fingers, and the response hand was counterbalanced across participants. Recognition testing occurred after a short distraction break to ensure retrieval based on long-term memory. Each participant viewed all 200 scenes from the encoding phase in a different random order, randomly interspersed with 100 similar but new scenes. The total pool of 300 scenes was randomly assigned into 3 groups of 100. Which of these 3 groups formed the old and new scenes for a particular participant was systematically counterbalanced across participants by rotation.

A short practice session with both encoding and recognition test phases was provided for each participant, rendering the experiment itself an intentional encoding paradigm, because participants knew during encoding that they would later be tested. Nevertheless, the focus of the encoding task was on deciding whether each depicted scene was indoors or outdoors.

In the recognition test, each scene was first shown for 1.25 s, then a 6-point scale was superimposed on the scene for 2.8 s, along which a marker was moved by pressing one of two keyboard buttons, with the index finger of each hand, to indicate degrees of confidence as to whether the scene was old or new (direction and response hand counterbalanced across participants). A fixation cross then appeared, jittered between 0.75 and 1.25 s. Behavioral data from the test phase are shown in *Table 1*. The data presented here were collapsed across the three scale points indicating 'old', and the three scale points indicating 'new' at test, to obtain binary 'old'/'new' judgments. All eight participants showed a greater percentage of hits (correct 'old' responses to old scenes) than of false alarms (incorrect 'old' responses to new scenes), demonstrating that they had formed memories for scenes from the encoding phase. The encoding data (electrophysiological data and response times) were then sorted, according to test phase responses, into epochs with successful encoding (later correct 'old' judgments to old scenes at test, hits) and epochs with unsuccessful encoding (later incorrect 'new' judgments to old scenes at test, misses) (*Paller and Wagner, 2002*).

## Data pre-processing

The electrophysiological encoding data were segmented into epochs 1 s pre-stimulus (i.e., scene presentation at encoding) to 2 s post-stimulus, because work on memory encoding measuring ERPs and theta/gamma oscillations has demonstrated differences in electrophysiological activity related to later memory up to 1.5 s post-stimulus, generally peaking at around 1 s after stimulus presentation (*Paller et al., 1987*; *Fell et al., 2001*; *Schott et al., 2002*; *Sederberg et al., 2003*; *Osipova et al., 2006*; *Lega et al., 2012*; *Long et al., 2014*). Data were also recorded during retrieval, but we focus on the encoding data here. Note that the time axes in all figures (except in *Figure 3—figure supplement 2*, where time is arbitrary) are set such that the stimulus was shown at time = 0 s.

Recording of intracranial signals was performed at the bedside in the few days following electrode implantation, before the electrodes were attached to a stimulator under the skin over the chest wall for epilepsy treatment, in a second operation approximately 1 week later. No seizures took place during the testing sessions, and all patients were fully alert and cooperative throughout. Epochs were individually visually inspected and cleaned of ocular and other artifacts using temporal-decorrelation-separation independent component analysis (*Sweeney-Reed et al., 2012*). Spikes and spike-waves were also removed using this approach, to maximize the number of epochs for analysis. We deemed the differences in electrophysiological activity detected between epochs recorded during successful compared with unsuccessful encoding in our simple task to be unlikely to result from global attentional fluctuations due to epileptiform activity for several reasons. Firstly, accuracy was close to ceiling for the indoor vs outdoor judgment during encoding. Secondly, there was no difference in response times during encoding for successful compared with unsuccessful memory formation. Thirdly, successful and unsuccessful encoding of successive scenes in the series showed no sequential dependency. Furthermore, while the cortical site of the epileptic focus differed across participants (*Table 2*), the

**Table 2.** Clinical information

| Pt | M/F | Age at surgery | Age at first seizure | Epilepsy syndrome | Seizure origin | Current medication |
|---|---|---|---|---|---|---|
| 1 | F | 42 | 13 | PLE | bilateral | LTG 200 mg |
| | | | | | | LCM 400 mg |
| 2 | F | 52 | 33 | TLE | bilateral | LCM 400 mg |
| | | | | | | LTG 200 mg |
| 3 | M | 34 | 26 | TLE | right | LEV 4000 mg |
| | | | | | | ESL 1200 mg |
| 4 | F | 29 | 16 | TLE | bilateral | LTG 400 mg |
| | | | | | | RTG 600 mg |
| 5 | M | 39 | 9 | TLE | left | LTG 400 mg |
| | | | | | | LCM 400 mg |
| 6 | M | 32 | 1 | TLE | frontal and right temporal | STP 4500 mg |
| | | | | | | OXC 900 mg |
| | | | | | | CLB 5 mg |
| 7 | M | 41 | 29 | FLE | bilateral | LTG 400 mg |
| | | | | | | ZNS 400 mg |
| 8 | F | 44 | 14 | TLE | left | CBZ 1200 mg |

Pt = participant. M/F = male/female. PLE = parietal lobe epilepsy. TLE = temporal lobe epilepsy. FLE = frontal lobe epilepsy. LTG = lamotrigine. LCM = lacosamide. LEV = levetiracetam. ESL = esilcarbazepine. RTG = retigabine. STP = striripentol. OXC = oxcarbazepine. CLB = clobazepam. ZNS = zonegran. CBZ = carbamazepine.

reported results were highly consistent across the group (*Figure 2E–F*, *Figure 2—figure supplements 2, 6–7*). Spikes or spike-waves were visible in the data from three of the eight participants (Participant 2: 10.3% during successful and 7.3% during unsuccessful encoding; Participant 6: 29.9% and 24.7% respectively; Participant 7: 2.7% and 5.4% respectively). Across these three participants, the mean difference in these percentages between successful and unsuccessful encoding was only 1.83% (i.e., slightly greater for successful encoding), suggesting that concerns about spikes and spike-waves being confounded with the two data categories of interest are minimal. Nevertheless, the analyses were also performed excluding epochs with spikes and spike-waves, and the findings reported remained statistically significant.

## Phase synchrony

Corticothalamic phase synchrony was calculated following wavelet time-frequency decomposition of each epoch, using 6-cycle Morlet wavelets, yielding 57 logarithmically spaced frequencies between 1 and 100 Hz. A logarithmic scale was used to take account of the frequency resolution of the Morlet wavelet (*Düzel et al., 2005*), such that wavelet spacing became sparser as frequency increased. After wavelet transformation, a phase series was extracted for each epoch, and the phase differences between scalp and thalamic channels were calculated (for details of this well known method (see *Lachaux et al., 1999* and *Sweeney-Reed et al., 2012*). Phase-locking values (PLVs) could vary between 0 (no PL) and 1 (complete PL). In order to enhance time resolution (at the expense of frequency resolution), we also applied 1-cycle wavelets, confirming that the synchrony that we report (*Figure 2*, *Figure 2—figure supplement 1*) took place post-stimulus (*Figure 2—figure supplement 5*). Statistical analysis thus focused on the 2 s post-stimulus period of 1024 time-points. It should be noted that the calculation of PLVs may be influenced by a common reference (*Vinck et al., 2011*), and we addressed this issue by using bipolar referencing in our thalamic recordings (see also *Staudigl et al., 2012*). Volume conduction may also influence PLVs (*Vinck et al., 2011*), but in the present study, phase synchrony is calculated between the frontal cortex and the RATN, whose spatial separation should exclude this influence.

## Statistical analysis

All statistical tests were two-tailed. After PLVs were calculated for each epoch in each category (successful encoding and unsuccessful encoding), permutation tests, which are conservative in that they do not make parametric assumptions, were initially used to evaluate the significance of differences between successful and unsuccessful encoding for each pixel within the 2-dimensional time-frequency space of 57 frequencies (0–100 Hz) and 1024 post-stimulus time-points (0–2 s). A mean PLV was calculated for successful and for unsuccessful encoding for each of the eight participants, and the 16 PLVs were pooled, reassigned to two artificial categories 1000 times, and a PLV difference between categories calculated, in order to obtain a two-tailed error distribution of differences against which the observed mean difference between successful and unsuccessful encoding was tested. A statistical comparison of mean PLVs across successful and unsuccessful encoding necessitates an equal number of epochs per category, so that, prior to calculating the mean PLVs for each participant, epochs were randomly selected from the larger category to match the size of the smaller (*Table 1*). *Figure 2A–C* shows the mean PLVs calculated from the equal number of epochs per category that were included in the statistical analysis. *Figure 2D* shows the results of the group permutation tests.

We also confirmed the key corticothalamic synchrony findings with paired T-tests (with mean PLVs calculated for each participant as just described for permutation testing). The assumptions of paired T-tests were satisfied (i.e., an approximately normal distribution of the differences between PLVs for successful and unsuccessful encoding across participants, and a positive correlation between PLVs for successful and unsuccessful encoding across participants). Mean PLVs differed significantly, in a pattern similar to that found using the permutation tests (*Figure 2—figure supplement 1*). All paired T-tests had 7° of freedom except where otherwise noted.

The electrophysiological literature on memory formation clearly suggests a focus on theta (4–8 Hz) oscillations, a range covered by 14 frequencies (i.e., wavelets). However, in view of the scarcity and novelty of memory-related intrathalamic data in humans, a highly conservative approach to evaluating these observed uncorrected p values was taken. For the permutation tests, a false discovery rate correction (*Canolty et al., 2006*) was applied for the 57 frequency (0–100 Hz) and 1024 time-point (0–2 s) comparisons, which was especially conservative given the dependency between adjacent time and frequency points (*Figure 2—figure supplement 1*). For the T-tests, a cluster-size permutation test (*Maris and Oostenveld, 2007*) was performed, in which a cluster was defined as adjacent significant (criterion: p = 0.05 by paired T-test) time-frequency points. Mean PLV matrices for each participant for each condition were randomly assigned to two groups 1000 times, and T-tests were performed across time and frequency for each permutation. T-test outcomes were rendered binary (1 = significant, 0 = nonsignificant) for each of the 57 frequencies and 1024 time-points, and the maximum cluster size (as defined above) emerging randomly in each iteration was calculated in order to provide a distribution against which to determine the significance of the observed cluster size. The same approach was taken for cluster-size significance in the cross-frequency coupling analysis described below. Finally, we evaluated the consistency of the results across the eight participants by applying a nonparametric (and thus conservative) Wilcoxon test of the differences between theta (5.2 Hz) PLVs for successful and unsuccessful encoding in the 0.5 to 1.5 s time-window in which theta differences had been revealed by the above-described conservative methods (see also *Figure 2*, *Figure 2—figure supplement 2*).

The significance of the difference between PLVs for successful vs unsuccessful encoding was additionally calculated on an individual case basis for Participants 1 and 2, because they were the only participants with parietal electrodes. Participant 1 had only 26 unsuccessful encoding epochs, so 26 successful encoding epochs were randomly selected from the 74 available and the mean PLV calculated. In order to use all available data to calculate the PLV difference between successful and unsuccessful encoding, this selection of 26 epochs and mean PLV calculation was carried out 200 times and an overall observed mean PLV was then calculated for successful encoding. For significance testing by permutation tests, this difference was compared to an error distribution, which was obtained by randomly assigning 52 epochs, irrespective of encoding success, to two artificial categories 1000 times (each time randomly selecting 26 of the 74 successful encoding epochs, and using all 26 unsuccessful encoding epochs) and calculating the mean PLV difference each time. The observed mean PLV difference was compared to this error distribution. A test with criterion p = 0.05 was applied (*Figure 2—figure supplement 6*, right). For descriptive purposes, the mean PLV across all 74 epochs of successful encoding is shown (*Figure 2—figure supplement 6*, left). Permutation of epochs was also performed

for Participant 2, with all 68 unsuccessful encoding epochs being used, and 68 encoding epochs being randomly selected from the 132 successful encoding epochs. The same procedure was then followed, except that given the larger number of trials in the smaller category compared with Participant 1, the observed mean PLV for successful encoding was calculated only once. For descriptive purposes, the mean PLV across all 132 epochs of successful encoding is shown (*Figure 2—figure supplement 7*, left).

## Cross-frequency coupling

Cross-frequency coupling (CFC) was calculated as per the work of *Canolty et al. (2006)* and *Axmacher et al. (2010)*. The frontal and RATN signals were wavelet transformed, then the theta phase and the gamma amplitude time series were extracted from the frontal and thalamic signals for each epoch. New complex signals were created by combining phases (ranging from 4–16 Hz, to include frequencies in the theta and alpha range) with gamma amplitudes (30–256 Hz) from the channels between which coupling was assessed, and the complex signals were averaged over time for each epoch. The phase was then extracted from the new complex value for each epoch by taking the arctangent of the imaginary divided by the real part. The average complex value across epochs was then also calculated, and from this value, the modulatory phase was determined by taking the arctangent of the imaginary over real parts, thus quantifying the average phase of the lower frequency oscillation at which the amplitude of the high frequency oscillation was highest. The theta amplitudes were then shifted by minus modulatory phase, and the correlation coefficient (CC) between the shifted theta oscillations and the gamma amplitudes was found. The CC was Fisher-Z transformed, then a mean was taken over epochs for each participant to provide the modulation index, separately for successful and unsuccessful encoding. Paired T-tests comparing successful with unsuccessful encoding were then performed for each pixel within the 37 theta-phase frequency by 21 gamma-amplitude frequency matrix, comparing the degree of theta-phase with gamma-amplitude coupling. The resulting p values, as shown in *Figure 3*, were then subjected to a cluster-based correction via permutation testing as described above for synchrony differences.

To illustrate the coupling in a single participant, the frontal and RATN theta phases and RATN gamma power are shown for Participant 7 in *Figure 3—figure supplement 2*, averaged across epochs during successful encoding. Note that to enable comparison across frequencies, the temporal mean was subtracted from the power values, and they were then divided by the temporal standard deviation of power (*Canolty et al., 2006*). Gamma power and theta troughs were aligned to the first theta trough in the 0.5 to 1.5 s post-stimulus window for each epoch (*Staudigl et al., 2012*).

The absence of an encoding-related difference in frontal CFC between upper theta peaks and gamma amplitudes is consistent with working memory findings (*Axmacher et al., 2010*). In order to confirm the presence of this CFC for both successful and unsuccessful encoding, levels of CFC were also compared with a distribution of CFC indices generated from 1000 phase-scattered surrogate data sets.

## Granger causality

Granger causality (GC) was used to investigate information flow direction (*Seth, 2010*; *Staudigl et al., 2012*) at time-frequency locations (4–12 Hz; 0.5–1.5 s) encompassing the significantly greater theta phase synchrony during successful compared with unsuccessful encoding. GC uses multivariate autoregressive modeling to ascertain whether time series A may be more accurately predicted from time series B, with a certain time lag, than B from A. If incorporating values from B in the regression of A allows better prediction of A than vice versa, B is said to influence A. The data were first detrended and rendered zero mean across epochs to remove nonstationarity. The variances of the prediction errors of the autoregressive models were then used to assess likely information flow direction. The Akaike and Bayesian information criteria (AIC, BIC) were employed to calculate a model order balancing overparameterization and adequate spectral resolution (*Seth, 2010*; *Staudigl et al., 2012*). When a minimum is not reached and the BIC/AIC does not show substantial decreases at higher orders, as here, model orders for EEG are usually chosen over a range (*Brovelli et al., 2004*; *Staudigl et al., 2012*). We applied model orders from 26 to 36, identifying a peak GC corresponding to a third of a theta cycle. Model coefficients were interpreted in the frequency domain. Significance of the difference between frontal-RATN and RATN-frontal GC in successful and in unsuccessful encoding was assessed by paired T-tests.

The flow of time is commonly used to make inferences from time series data regarding directional causal influences (*Bollimunta et al., 2008*), and indeed GC is commonly equated with directions of information flow in neural circuits (*Ding et al., 2006*; *Bollimunta et al., 2008*; *Anderson et al., 2010*).

The model order specifies the number of time lagged observations made (*Bressler and Seth, 2011*), and the order at which GC peaks provides an indication of how long into the past one signal provides information about subsequent activity in the other signal. We interpreted the model order corresponding with peak causality as providing an indication of the delay in transfer of information (*Staudigl et al., 2012*), which has been measured directly between the frontal cortex and hippocampus in rats and found to be consistent with the delays suggested by the present data (*Siapas et al., 2005*; *Benchenane et al., 2010*).

## Acknowledgements

The authors would like to thank Michael Rugg for discussion concerning data analysis, and Erik Dower for discussion concerning analog signals. This research was supported by German Research Foundation (DFG) Grants 1847/1-1 and SFB779 TPA10N (AR-K), and by the Humboldt Foundation (AR-K).

## Additional information

### Funding

| Funder | Grant reference number | Author |
| --- | --- | --- |
| Alexander von Humboldt-Stiftung | | Alan Richardson-Klavehn |
| Deutsche Forschungsgemeinschaft | 1847/1-1, SFB779 TPA10N | Alan Richardson-Klavehn |

The funders had no role in study design, data collection and interpretation, or the decision to submit the work for publication.

### Author contributions

CMS-R, Conceived and designed the research, Prepared the experiment, Analyzed and interpreted the behavioral and electrophysiological data, including writing data-analysis programs, Drafted and revised the article; TZ, Acquired the behavioral and electrophysiological data, Revised the article; JV, Acquired data: selected and provided the technical facilities used for the human stereotactic neurosurgery and intracranial recording, recommended and supervised patient treatment with deep brain stimulation, surgically implanted the intracranial electrodes, Analyzed and interpreted data: localized the intracranial electrodes, Revised the article; FCS, Acquired data: diagnosed the patients, recommended and supervised patient treatment with deep brain stimulation, Revised the article; LB, KK, CE, Analyzed and interpreted data: localized the intracranial electrodes, Revised the article; HH, H-JH, Acquired data: selected and provided the technical facilities used for the human stereotactic neurosurgery and intracranial recording, Analyzed and interpreted data: commented on the data analysis, Revised the article; RTK, Analyzed and interpreted data: advised on the data analysis, Advised on and revised the article; AR-K, Conceived and designed the research, Analyzed and interpreted the behavioral and electrophysiological data, Drafted and revised the article

### Author ORCIDs

Catherine M Sweeney-Reed, (iD) http://orcid.org/0000-0002-3684-1245

### Ethics

Human subjects: The measurements were approved by the Ethics Commission of the Medical Faculty of the Otto-von-Guericke University, Magdeburg, and all participants gave written informed consent.

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
