## [Decision Letter]

Thank you for sending your work entitled “Corticothalamic phase synchrony and cross-frequency coupling predict human memory formation” for consideration at *eLife*. Your article has been favorably evaluated by Eve Marder (Senior editor), a Reviewing editor, and 3 reviewers.

The following individuals responsible for the peer review of your submission have agreed to reveal their identity: Howard Eichenbaum (Reviewing editor) and Guillén Fernández (one of the three peer reviewers).

The Reviewing editor and the reviewers discussed their comments before we reached this decision, and the Reviewing editor has assembled the following comments to help you prepare a revised submission.

This is a timely and valuable study probing the neural correlates of memory formation, presenting rare data on intracranial recordings from human patients focusing on the thalamus, with simultaneous cortical EEG from frontal and parietal sites. The authors assessed phase synchronization and cross frequency coupling in recordings from eight patients with epilepsy, who performed an intentional picture encoding task followed by a recognition memory test. They confirmed the hypothesis that the interaction between the anterior thalamus and frontal cortex is related to successful memory formation and based on cross-region theta synchronization and local theta-gamma coupling. The paper is clearly and concisely written but each of the reviewers had several concerns that should be addressed in a revision.

*Reviewer #1*:

1) The manuscript says very little about the patients and their epilepsy (this should be added). Related to that, the reader does not know whether the data (and performance) is contaminated by epileptic activity. Spike and spike-waves are able to affect cognition, including the ability to encode, and the analyses done by the authors. Thus, they have to make clear by appropriate correction or better trial exclusion that the recording epochs used for analysis are not affected by interictal or even ictal epileptic activity.

2) The task for the patients was to recognize a large number of photographs making it likely to be solved, at least to a substantial degree, by recognition judgments based on a feeling of familiarity. In other words, this test is not a clean episodic memory task as framed by the authors and thus, this is a limitation that should be discussed.

3) The authors did some analysis in an attempt to exclude attentional explanations for their findings. I think that this is neither necessary nor convincing. Reaction time differences in the simple orientation task (indoor/outdoor decision) might be related to modulations of attention, but they are dominated by stimulus features. The decision is just more difficult for some items than others. Also analyzing the probability of sequential encoding is not convincing. This analysis might track slowly modulated changes in attentional states, but it fails to detect trial-by-trial fluctuations of attention, which might be most relevant for the subsequent memory effect as assessed here. However, the subsequent memory paradigm depends on fluctuations in the probability to encode a series of items. If you would theoretically be able to align perfectly attention, stimuli, emotional valence and all other factors modulating the likelihood to encode an item successfully, a given subject would encode either all items or none.

Somewhere, the subsequent memory design needs fluctuations in these factors. If the subject is able to perform the orientation task correctly (and above chance level), we can assume that he/she has processed the items attentively, further fluctuations of attention are most likely present and related to memory formation success, but this is no problem because these fluctuations are needed for this study design. To dissociate attentional from mnemonic components, one would for instance need a factorial design modulating memory formation and attention; very difficult to come up with a clever design (see Rees et al. Science 1999 for an attempt). To my opinion, the authors can just omit this attentional topic in their Results and Discussion section.

4) The authors are strict in localizing their effect in the anterior thalamic nucleus and distinguish this location from recordings in the dorsomedial thalamic nuclei. The data presented is not sufficient to support this specificity. Also the framing is too focused. For instance, they ignore the nucleus reunions, a most relevant thalamic relay station interacting with the prefrontal cortex and medial temporal lobe during specific mnemonic processes (e.g., Xu and Südhof, Science 2013). Given the connectivity of this nucleus, it is way better suited to support memory formation by connecting to the (medial) prefrontal cortex. I suggest to present the electrode localization in more detail and to broaden the framing.

*Reviewer #2*:

1) A fairly obvious question: what happens during retrieval? Were the data collected? Are there plans for describing them in a separate publication? That would be fine, but it would be good to know.

2) Could you please show raw power data for the successful/unsuccessful conditions? It would be important to know whether power also increases in the successful condition, and, in that case, whether the increased PLV computed could be ascribed to improved statistical power. Measures that take power into account as Martin Vinck's WPLI may be considered here.

3) The authors state “The narrower gamma range involved during successful encoding may be interpreted as selection, via corticothalamic coupling, of RATN neural assemblies specific to a particular memory trace” I'm not sure I follow the logic behind this statement, could you please explain?

4) The authors also state “Such a delay is broadly consistent with hippocampal-to-medial-prefrontal-cortex delays reported during theta phase synchrony in rats (5; 58)”. Why does it make sense to compare to HC-PFC interactions here? In the present case the cortex is leading, in the former case is lagging.

*Reviewer #3*:

1) I am not certain that this study is actually testing episodic memory. The researchers use a recognition memory paradigm that includes a confidence measurement. One might argue that highly confident measurements are episodic-type retrieval more than recognition but in the analysis they appear to collapse all confidence measurements into two different categories. Given the available data, I assume it is difficult to disambiguate highly confident from only partially confident measurements, but I would prefer to see something that attempts to look more strongly at the “recall” type encoding events rather than those that are simply recognition events. Further, in a recognition memory paradigm, activity at retrieval is probably as important as activity during encoding for determining success and an analysis focusing on this epoch would be of interest. Specifically, examining activity time-locked to retrieval item presentation would be interesting. I think the study is still noteworthy, even if it assesses recognition memory and not episodic memory, but they authors may wish to re-cast their findings somewhat.

2) I am not certain of the region-specificity of the memory-related thalamic-cortical effect. The anterior thalamic nucleus was selected as a target for epilepsy because of wide cortical projections, a “way-station” between localized onset and generalization. Clinically, part of ATN electrode placement includes proving that stimulation elicits a driving response in multiple cortical locations (Lega, B. C., Halpern, C. H., Jaggi, J. L., & Baltuch, G. H. (2010). Deep brain stimulation in the treatment of refractory epilepsy: update on current data and future directions. Neurobiology of disease, 38(3), 354-360.) To my eyes, in the supplementary figures, the parietal response is more robust than the examples of a frontal response that they show. Certainly, the memory-relevant thalamic connections may include multiple cortical areas, but if the memory effect does not have any region-specificity it may indicate orienting or attention, etc. more than a memory specific effect. Obtaining more scalp electrode data in these patients may be possible for future cases (dressing issue can be overcome by placing scalp leads under sterile conditions in the OR), but if the authors have any other contacts to report or can compare the frontal versus parietal activity this may help. They can also maybe differentiate frontal pole from FCz activity in the aggregated data. This would apply both to the synchrony analysis and the CFC analysis.

3) I have a few concerns about the synchrony analysis. The clustering analysis shows significant synchrony in short time periods within the .5-1.5 second epoch, but for a 5 Hz oscillation I would expect something like .5-1 seconds of continuously significant synchrony difference via the PLV value across subjects. But when they compare the PLV value for the entire time window and compare across the subjects, the effect (p = 0.038) may not survive a correction for multiple comparisons across all of the different tests they performed. I would prefer to see more analyses that look across the relevant time window, long enough to account for a few cycles of the oscillation. Similarly, while the PLV difference is significant, I would like to see something (maybe I missed it) showing the level of synchrony by itself is significant for the same relevant, contiguous time epoch, at least for the putatively higher-synchrony condition (successful encoding). I assume the authors did this following the LaChaux methods but a plot showing the time windows that are significantly synchronous would be helpful.

4) I would prefer to see an analysis that includes a closer look at the different frequency components. Differential coupling of the higher-frequency (alpha range) component versus the lower frequency (4-8 Hz) component (for example frontal pole versus dorsal frontal versus parietal) would help establish a memory-specific effect for the theta component. Similarly, I would prefer to see the effects into the delta frequency range (2-4 Hz) given human data suggesting a memory-relevant effect in this frequency range (Lega et al., Slow-Theta-to-Gamma Phase-Amplitude Coupling in Human Hippocampus Supports the Formation of New Episodic Memories, Cereb Cortex. 2014 Oct 14. pii: bhu232, Watrous, Andrew J., Itzhak Fried, and Arne D. Ekstrom. “Behavioral correlates of human hippocampal delta and theta oscillations during navigation.” Journal of Neurophysiology 105.4 (2011): 1747-1755).

---

## [Author Response]

Reviewer #1:

*1) The manuscript says very little about the patients and their epilepsy (this should be added). Related to that, the reader does not know whether the data (and performance) is contaminated by epileptic activity. Spike and spike-waves are able to affect cognition, including the ability to encode, and the analyses done by the authors. Thus, they have to make clear by appropriate correction or better trial exclusion that the recording epochs used for analysis are not affected by interictal or even ictal epileptic activity*.

We have added a table (Table 2) providing details about the epilepsy diagnosis for each patient.

We have also added more information to the first paragraph of the 'Participants, and intracranial and scalp recording' section of the 'Materials and methods' section, so the first sentence is now as follows:

“Intrathalamic data were recorded from 1.5 mm platinum-iridium electrodes implanted bilaterally (4 contacts each) in the thalamus for stimulation therapy for multiple pharmacoresistant focal epilepsy in 8 adult participants, all of whom were not suitable candidates for resective surgery.”

Recording of intracranial signals was only possible at the patient's bedside during the few days directly following surgery, limiting the time available for such studies, because the electrodes were subsequently attached to a stimulator under the skin during a second operation. To maximize the number of epochs for analysis, for the main analyses reported here, we corrected for artifacts instead of excluding trials. We have re-analyzed the data excluding all trials containing spikes or spike-waves, and the findings reported as significant in the main analyses remain significant. We have correspondingly altered the second paragraph of the 'Data pre-processing' section of the 'Materials and methods' section to the following:

“Recording of intracranial signals was performed at the bedside in the few days following electrode implantation, before the electrodes were attached to a stimulator under the skin over the chest wall for epilepsy treatment, in a second operation approximately one week later. […] Nevertheless, the analyses were also performed excluding epochs with spikes and spike-waves, and the findings reported remained statistically significant.”

*2) The task for the patients was to recognize a large number of photographs making it likely to be solved, at least to a substantial degree, by recognition judgments based on a feeling of familiarity. In other words, this test is not a clean episodic memory task as framed by the authors and thus, this is a limitation that should be discussed*.

We agree with the reviewer that familiarity cannot be excluded as playing a role in our retrieval task. Given the novelty and scarcity of human intrathalamic recordings, we chose to implement a simple paradigm, which we were confident the patients could successfully perform, given that the only possible time to record such data was during the few days following intracranial surgery. Furthermore, in order to maximize trial numbers for the synchrony analyses, we collapsed the data across the levels of certainty used by the participants for their recognition judgments at test, meaning that we cannot conclusively link our results to recollection separately from familiarity. We do not regard this fact as undermining the novelty and importance of our data, because the ATN has been linked through human lesion data to the formation of episodic memories (64; 24). Nevertheless, we do not wish to make strong claims in this regard, and have removed the word 'episodic' from relevant places in the manuscript and have added the following:

“The encoding and retrieval tasks were chosen for their simplicity, because recording was only possible at the bedside in the few days following intracranial surgery, and because we did not wish to lose data owing to failure of these rare participants to succeed at the tasks. […] Despite this lack of behavioral/psychological resolution, our data nevertheless provide novel evidence concerning the real-time role of the human ATN in memory formation.”

Please also see our response to a similar point raised by Reviewer 3, point 1.

*3) The authors did some analysis in an attempt to exclude attentional explanations for their findings. I think that this is neither necessary nor convincing. Reaction time differences in the simple orientation task (indoor/outdoor decision) might be related to modulations of attention, but they are dominated by stimulus features. The decision is just more difficult for some items than others. Also analyzing the probability of sequential encoding is not convincing. This analysis might track slowly modulated changes in attentional states, but it fails to detect trial-by-trial fluctuations of attention, which might be most relevant for the subsequent memory effect as assessed here. However, the subsequent memory paradigm depends on fluctuations in the probability to encode a series of items. If you would theoretically be able to align perfectly attention, stimuli, emotional valence and all other factors modulating the likelihood to encode an item successfully, a given subject would encode either all items or none*.

*Somewhere, the subsequent memory design needs fluctuations in these factors. If the subject is able to perform the orientation task correctly (and above chance level), we can assume that he/she has processed the items attentively, further fluctuations of attention are most likely present and related to memory formation success, but this is no problem because these fluctuations are needed for this study design. To dissociate attentional from mnemonic components, one would for instance need a factorial design modulating memory formation and attention; very difficult to come up with a clever design (see Rees et al. Science 1999 for an attempt). To my opinion, the authors can just omit this attentional topic in their Results and Discussion section*.

We thank the reviewer for making these points, and agree completely that variation in attention and other cognitive variables across items at encoding is necessary in order actually to enable a comparison of successful with unsuccessful memory encoding. Reviewer #3 raises the question, however, as to whether our findings reflect attentional processes, and Reviewer #1 (point 1) previously raises the question of whether attentional lapses due to epileptiform activity might be responsible for failures to encode, so we have retained the attention-related behavioral analyses in the Results section, and we have added the following to the Discussion section to incorporate these points made by Reviewer #1:

“We note, however, that variability in factors such as attention and the emotional valence of stimuli are fundamental to the study design in subsequent memory paradigms, affecting the probability of successfully encoding each item and thus enabling comparison of successful with unsuccessful encoding. Indeed, the early upper theta and alpha synchrony immediately following stimulus presentation during successful memory encoding is likely to reflect enhanced attention and perception (16), with the later theta synchrony timing fitting well with previous findings relating to memory encoding (53; 23).”

*4) The authors are strict in localizing their effect in the anterior thalamic nucleus and distinguish this location from recordings in the dorsomedial thalamic nuclei. The data presented is not sufficient to support this specificity. Also the framing is too focused. For instance, they ignore the nucleus reunions, a most relevant thalamic relay station interacting with the prefrontal cortex and medial temporal lobe during specific mnemonic processes (e.g., Xu and Südhof, Science 2013). Given the connectivity of this nucleus, it is way better suited to support memory formation by connecting to the (medial) prefrontal cortex. I suggest to present the electrode localization in more detail and to broaden the framing*.

The significant theta synchrony difference between successful and unsuccessful encoding which we identified in the ATN was not detected in the DMTN. Moreover, this synchrony difference in the ATN was greater than that in the DMTN on direct comparison. We do not, however, intend to suggest that these findings mean that the ATN is the only thalamic nucleus to have a role in memory processing. Indeed published electrophysiological data from our research group already show a role for the DMTN in retrieval processing in humans (60). We thank the reviewer for the suggestion of broadening the framing to comment on the importance of other thalamic nuclei in different aspects of memory processing.

The nucleus reuniens has indeed been demonstrated in rats to play a role in memory, specifically in fear conditioning and memory generalization (73), with prefrontal cortical and hippocampal connectivity supporting a regulatory role in memory processing (67). Recently this nucleus has also been found to play a role in spatial processing, which may complement hippocampal spatial processing (25). We do not have recordings from the nucleus reuniens because the electrode placement was determined for clinical purposes. Placement of electrodes in the ATN and DMTN provided a rare opportunity to record electrophysiological data from the human thalamus, and we investigated memory function, because human lesion studies and animal studies suggest that these nuclei play a role in memory processing (24; 64; 2; 1; 66; 60). The location of the nucleus reuniens relative to the ATN and DMTN is such that stereotactic placement of the electrodes in the ATN and DMTN precludes inadvertent placement in the nucleus reuniens (distance between ATN-/DMTN-targets and N. reuniens 10-15 mm), and indeed post-operative imaging was used to confirm correct electrode placement. Furthermore, the connectivity of the nucleus reuniens does not include direct communication with the ATN or DMTN (Cassel et al. 2013). Indeed it has been suggested that the nucleus reuniens is involved in a separate memory processing system to that incorporating the ATN and DMTN (39).

We have added the following to the manuscript to broaden the framing as suggested:

“It should also be noted that memory processing may be divided into different subsystems, with different components of memory processing involving different thalamic nuclei (39). For example, whereas human lesion and animal studies suggest a regulatory role for the ATN in memory encoding (24; 64; 2; 1; 66) and for the DMTN in retrieval (64; 60), other thalamic nuclei have also been found to be involved in different aspects of memory processing. […] Our focus on the ATN and DMTN is based on the rare availability of human electrophysiological data from these sites, which was determined by clinical requirements, and is consistent with extant data concerning the amnesic effects of lesions in these thalamic areas in humans (64; 24).”

We have also added more detail to the 'Participants, and intracranial and scalp recording' part of the 'Materials and methods' section regarding localization of the electrodes:

“Placement of the thalamic electrodes was performed stereotactically. The angle of entry through the skull and the depth of each electrode was calculated based on MRI images of each patient's brain pre-operatively. An intra-operative X-ray and postoperative CT-scans were carried out in order to confirm correct localization of each electrode, by reference to the Schaltenbrand and Pick Atlases (52; 37).”

Reviewer #2:

*1) A fairly obvious question: what happens during retrieval? Were the data collected? Are there plans for describing them in a separate publication? That would be fine, but it would be good to know*.

The retrieval data were collected, and this question is indeed the subject of analyses in progress and the preparation of a separate manuscript. In this context, we note that encoding data are often reported alone ([46]; Gottlieb et al. 2012; [71]; [56]; [21]; [45]; [35]; [53]; [16]; [54]), and we hope not to delay communication of the present encoding findings. As the reviewer and editors will appreciate from the substantial number of figures needed to illustrate our encoding data alone, including the retrieval data would result in a publication that would be long and difficult to digest for a reader, thus undermining the impact of the principal points made here.

In order to address the question, we have added:

“Data were also recorded during retrieval, but we focus on the encoding data here.”

*2) Could you please show raw power data for the successful/unsuccessful conditions? It would be important to know whether power also increases in the successful condition, and, in that case, whether the increased PLV computed could be ascribed to improved statistical power. Measures that take power into account as Martin Vinck's WPLI may be considered here*.

The significance of the difference between power during successful compared with unsuccessful encoding is shown in Figure 2—figure supplement 8. We have now added panels showing the raw power data for successful compared with unsuccessful encoding frontally and in the RATN, as requested.

Given the absence of a significant difference in power, we chose to calculate phase-locking values, which measure timing separately from power. We considered application of the WPLI, but given that the power difference was not significant, and that the other concern WPLI was designed to address, namely volume conduction, is not likely to be a confounding factor in our analyses, given the spatial separation of the frontal cortex and the RATN, we believe the method we chose is appropriate. We have added the following to the manuscript to address this point:

“It should be noted that the calculation of PLVs may be influenced by a common reference (68), and we addressed this issue by using bipolar referencing in our thalamic recordings (see also [60]). Volume conduction may also influence PLVs (68), but in the present study, phase synchrony is calculated between the frontal cortex and the RATN, whose spatial separation should exclude this influence.”

*3) The authors state “The narrower gamma range involved during successful encoding may be interpreted as selection, via corticothalamic coupling, of RATN neural assemblies specific to a particular memory trace” I'm not sure I follow the logic behind this statement, could you please explain*?

We thank the reviewer for pointing out that this statement of ours was not clear. The narrower gamma range could be interpreted as reflecting the firing of fewer neural assemblies, that is, those pertaining to a particular memory trace. We have modified the sentence to clarify this point:

“Assuming that an assembly of synchronously firing neurons is associated with a particular memory trace, the narrower RATN gamma range coupled with theta phase during successful memory formation could be interpreted as reflecting firing of only relevant neural assemblies, thus reflecting neural specificity during encoding (16; 14; 54). We correspondingly postulate that the corticothalamic coupling may coordinate the firing of particular thalamic neural assemblies underpinning the memory to be encoded, facilitating synaptic strengthening and relevant memory formation.”

*4) The authors also state “Such a delay is broadly consistent with hippocampal-to-medial-prefrontal-cortex delays reported during theta phase synchrony in rats (*[5]*;*[58]*)”. Why does it make sense to compare to HC-PFC interactions here? In the present case the cortex is leading, in the former case is lagging*.

We agree and have eliminated this point from the manuscript.

Reviewer #3:

*1) I am not certain that this study is actually testing episodic memory. The researchers use a recognition memory paradigm that includes a confidence measurement. One might argue that highly confident measurements are episodic-type retrieval more than recognition but in the analysis they appear to collapse all confidence measurements into two different categories. Given the available data, I assume it is difficult to disambiguate highly confident from only partially confident measurements, but I would prefer to see something that attempts to look more strongly at the “recall” type encoding events rather than those that are simply recognition events. Further, in a recognition memory paradigm, activity at retrieval is probably as important as activity during encoding for determining success and an analysis focusing on this epoch would be of interest. Specifically, examining activity time-locked to retrieval item presentation would be interesting. I think the study is still noteworthy, even if it assesses recognition memory and not episodic memory, but they authors may wish to re-cast their findings somewhat*.

The reviewer is correct that separating responses according to confidence does not provide optimal trial numbers for statistical comparison of the electrophysiological data using synchrony measures. Given the novelty of the thalamic data, we elected to maximize the number of trials to maximize the power of the electrophysiological analyses, and our confidence in their reliability. As a result, as noted by the reviewer, we cannot disambiguate familiarity from recollection, and we have re-cast our findings accordingly in the manuscript. (Please see our response to Reviewer #1, point 2.) We intend to explore retrieval in subsequent work that is beyond the scope of the current manuscript for reasons also mentioned earlier. (Please see our response to Reviewer #2, point 1.)

*2) I am not certain of the region-specificity of the memory-related thalamic-cortical effect. The anterior thalamic nucleus was selected as a target for epilepsy because of wide cortical projections, a “way-station” between localized onset and generalization. Clinically, part of ATN electrode placement includes proving that stimulation elicits a driving response in multiple cortical locations (Lega, B. C., Halpern, C. H., Jaggi, J. L., & Baltuch, G. H. (2010). Deep brain stimulation in the treatment of refractory epilepsy: update on current data and future directions. Neurobiology of disease, 38(3), 354-360.) To my eyes, in the supplementary figures, the parietal response is more robust than the examples of a frontal response that they show. Certainly, the memory-relevant thalamic connections may include multiple cortical areas, but if the memory effect does not have any region-specificity it may indicate orienting or attention, etc. more than a memory specific effect. Obtaining more scalp electrode data in these patients may be possible for future cases (dressing issue can be overcome by placing scalp leads under sterile conditions in the OR), but if the authors have any other contacts to report or can compare the frontal versus parietal activity this may help. They can also maybe differentiate frontal pole from FCz activity in the aggregated data. This would apply both to the synchrony analysis and the CFC analysis*.

We thank the reviewer for this suggestion and now show the difference between PLVs during successful minus unsuccessful encoding as a mean across those participants with frontopolar electrodes, those with other frontal electrodes, and those with electrodes at Pz (Figure 2—figure supplement 9). While the synchrony pattern we report is discernible with all 3 cortical electrode placements, it is not identical. We also show the difference between frontal-RATN theta-gamma cross-frequency coupling involving the different frontal electrode placements. Again, the patterns are similar but not identical. We cannot, however, make strong claims about cortical specificity given the number of participants with each electrode placement. Participants 3-8 had only a single frontal electrode, at Fpz, and only participants 1 and 2 had other frontal and also parietal contacts, meaning that power is inadequate for a group statistical analysis to compare differing frontal and parietal synchrony directly.

We performed several additional analyses to address the question of the role of attention in our data, and note that attention plays a role in memory formation. (Please see point 3 by Reviewer #1.)

We have now addressed the issue of cortical specificity as follows, including adding Figure 2—figure supplement 9 and Figure 3—figure supplement 3:

“The anterior thalamic nucleus is the target of stimulation in the treatment of focal epilepsy on the basis that seizure activity starting focally is propagated through this site to widespread cortical areas (32). Regional cortical specificity in frontal-RATN synchrony would support our argument that the differences that we have identified are memory-specific. […] We note also, though, that both frontal and parietal cortices are well-recognized as being involved in memory processing (71; 63; 51; 61; 21; 45).”

*3) I have a few concerns about the synchrony analysis. The clustering analysis shows significant synchrony in short time periods within the .5-1.5 second epoch, but for a 5 Hz oscillation I would expect something like .5-1 seconds of continuously significant synchrony difference via the PLV value across subjects. But when they compare the PLV value for the entire time window and compare across the subjects, the effect (p = 0.038) may not survive a correction for multiple comparisons across all of the different tests they performed. I would prefer to see more analyses that look across the relevant time window, long enough to account for a few cycles of the oscillation. Similarly, while the PLV difference is significant, I would like to see something (maybe I missed it) showing the level of synchrony by itself is significant for the same relevant, contiguous time epoch, at least for the putatively higher-synchrony condition (successful encoding). I assume the authors did this following the LaChaux methods but a plot showing the time windows that are significantly synchronous would be helpful*.

We performed a cluster analysis in order to account for multiple comparisons in terms of time and frequency bins. The literature suggests that successful encoding might involve theta synchrony, because this frequency dominates in the hippocampus, whose role in memory processing is well-established, is the frequency at which ATN oscillations have been identified, and is the frequency implicated in ATN-hippocampal communication (66; 2). Moreover, human lesion and animal studies suggest a role for the ATN in successful memory formation (24; 64; 2). We provided analyses involving the DMTN, because data from this nucleus were also available, and such analysis supports the proposal that the theta synchrony is specific to the ATN, as predicted. We chose to display time-frequency plots that provide information regarding other frequencies in order to present a complete picture, despite justification for a focus simply on theta. The reviewer is correct that we applied phase synchrony analysis as proposed by Lachaux et al. We have added a further figure supplement to Figure 2 to address the reviewer's request for a plot showing that theta synchrony against time during the 0.5-1.5 s time window. Compared with surrogate data (62), the PLV during successful encoding was significant (criterion *P* = 0.05) for 230 ms, which is more than a complete theta cycle at 5 Hz. The timescale is of the order of that over which synchrony is commonly detected (65).

We have added the following:

“The time course of theta synchrony is shown in Figure 2—figure supplement 3. Compared with surrogate data, the PLV was significant (criterion *P* = 0.05) for 230 ms, which is more than a complete theta cycle at 5 Hz. The timescale is of the order of that over which synchrony is commonly detected (65). Note from Figure 2 and Figure 2—figure supplement 2 that theta synchrony was greater during successful than unsuccessful encoding in all 8 participants from 1.25-1.5 s post-stimulus, and the appearance of two separate episodes at the group level is likely to have arisen due to inter-participant differences in the timing of the synchrony episode over the 1 s time window from 0.5-1.5 s post-stimulus.”

We have also added Figure 2—figure supplement 3.

*4) I would prefer to see an analysis that includes a closer look at the different frequency components. Differential coupling of the higher-frequency (alpha range) component versus the lower frequency (4-8 Hz) component (for example frontal pole versus dorsal frontal versus parietal) would help establish a memory-specific effect for the theta component. Similarly, I would prefer to see the effects into the delta frequency range (2-4 Hz) given human data suggesting a memory-relevant effect in this frequency range (Lega et al., Slow-Theta-to-Gamma Phase-Amplitude Coupling in Human Hippocampus Supports the Formation of New Episodic Memories, Cereb Cortex. 2014 Oct 14. pii: bhu232, Watrous, Andrew J., Itzhak Fried, and Arne D. Ekstrom. “Behavioral correlates of human hippocampal delta and theta oscillations during navigation.” Journal of Neurophysiology 105.4 (2011): 1747-1755)*.

Please see our response to point 2 regarding the separation of frontal sites. We have now carried out a comparison of theta synchrony with both alpha and beta, finding the effect to be specific to theta.

We did not find significant phase synchrony differences between successful and unsuccessful encoding in the delta frequency range, and we focused on the theta frequency range in the subsequent analyses due to its theoretical relevance in the ATN. We have now added this information to the Results and Discussion sections as follows:

“To further test frequency-specificity, a two-way repeated measures analysis of variance was applied to frontal-RATN PLVs obtained using theta (5.2 Hz) and beta (17.5 Hz) wavelets averaged from 0.5 to 1.5 s for each participant during successful and unsuccessful encoding. The mean PLVs showed a significant interaction between frequency and encoding success (*P* < 0.001). This interaction remained significant when taking adjacent theta (4.9 Hz and 5.5 Hz) and beta (16.5 Hz and 18.6 Hz) wavelets, and taking theta (5.2 Hz) and alpha (11.7 Hz) wavelets (all *P*s < 0.002). In all cases the advantage for successful over unsuccessful encoding for theta was greater than for alpha or beta, which showed negligible differences. The interaction was not significant when taking two theta wavelets (4.9 Hz and 5.2 Hz: *P* > 0.30; 5.2 Hz and 5.5 Hz: *P* > 0.60). No significant difference was observed between delta (2-4 Hz) phase synchrony during successful compared with unsuccessful encoding.”

“While oscillations in the delta frequency range have recently been found to show a memory-related difference in the hippocampus (31; 69), we did not detect a significant difference between corticothalamic delta phase synchrony during successful compared with unsuccessful encoding.”